# Diet Supplementation with Fish-Derived Extracts Suppresses Diabetes and Modulates Intestinal Microbiome in a Murine Model of Diet-Induced Obesity

**DOI:** 10.3390/md19050268

**Published:** 2021-05-11

**Authors:** Konstantinos Axarlis, Maria G. Daskalaki, Sofia Michailidou, Nikolais Androulaki, Antiopi Tsoureki, Evangelia Mouchtaropoulou, Ourania Kolliniati, Ioanna Lapi, Eirini Dermitzaki, Maria Venihaki, Katerina Kousoulaki, Anagnostis Argiriou, Zouhir El Marsni, Christos Tsatsanis

**Affiliations:** 1Laboratory of Clinical Chemistry, Medical School, University of Crete, 70013 Heraklion, Greece; molgrad392@edu.biology.uoc.gr (K.A.); m.daskalaki@med.uoc.gr (M.G.D.); chem1813@edu.chemistry.uoc.gr (N.A.); medp2011892@med.uoc.gr (O.K.); medp2011988@med.uoc.gr (I.L.); renaderm@med.uoc.gr (E.D.); venihaki@med.uoc.gr (M.V.); 2Institute of Molecular Biology and Biotechnology, FORTH, 71100 Heraklion, Greece; 3Institute of Applied Biosciences (INAB), CERTH, Thermi, 57001 Thessaloniki, Greece; sofia_micha28@certh.gr (S.M.); adatsoureki@certh.gr (A.T.); eva.mouchtaropoulou@certh.gr (E.M.); argiriou@certh.gr (A.A.); 4Department of Nutrition and Feed Technology, Nofima AS, 5141 Bergen, Norway; katerina.Kousoulaki@Nofima.no; 5Department of Food Science and Nutrition, University of the Aegean, Myrina, 81400 Lemnos, Greece; 6Seagarden AS, 4262 Avaldsnes, Norway

**Keywords:** obesity, type 2 diabetes, insulin resistance, fish extract, collagen, gut microbiome

## Abstract

Metabolic syndrome-related diseases affect millions of people worldwide. It is well established that changes in nutritional habits and lifestyle can improve or prevent metabolic-related pathologies such as type-2 diabetes and obesity. Previous reports have shown that nutritional supplements have the capacity to limit glucose intolerance and suppress diabetes development. In this study, we investigated the effect of dietary supplementation with fish-derived extracts on obesity and type 2 diabetes and their impact on gut microbial composition. We showed that nutritional supplements containing Fish Complex (FC), Fish Complex combined with Cod Powder (FC + CP), or Cod Powder combined with Collagen (CP + C) improved glucose intolerance, independent of abdominal fat accumulation, in a mouse model of diet-induced obesity and type 2 diabetes. In addition, collagen-containing supplements distinctly modulate the gut microbiome in high-fat induced obesity in mice. Our results suggest that fish-derived supplements suppress diet-induced type 2 diabetes, which may be partly mediated through changes in the gut microbiome. Thus, fish-derived supplements and particularly the ones containing fish collagen have potential beneficial properties as dietary supplements in managing type 2 diabetes and metabolic syndrome via modulation of the gut microbiome.

## 1. Introduction

Obesity and insulin resistance are central traits of the metabolic syndrome and contribute to serious, yet alarmingly frequent pathologies, such as type 2 diabetes, hypertension and cardiovascular disease [1,2]. Based on the World Health Organization’s official statistics, in 2016, more than 1.9 billion adults were overweight, of whom 650 million were obese. Obesity is defined as the abnormal or excessive fat accumulation which poses a risk to health, whereas insulin resistance is the inability of the body to utilize insulin efficiently [3]. In many cases, both are caused by lifestyle behaviors and, thus, are preventable. More specifically, dietary habits such as excess calorie or processed food consumption, and, importantly, lack of sufficient physical activity are fundamental to shaping these two detrimental conditions [4,5,6]. Other risk factors may include genetic predisposition, chronic inflammation, alcohol consumption, stress, age, medications, diseases, disturbed circadian rhythm, and microbiome content [1,2,7].

The gut microbiome’s contribution to the development of obesity has only recently been highlighted, while it possibly expands to other metabolic diseases too, such as type-2 diabetes. Intestinal microbes are in constant and complex interaction with the host, affecting its immune, endocrine and nervous system in addition to its metabolism. Dietary interventions have a considerable impact on the prevention, progression and treatment of obesity and insulin resistance [1,2,8]. Numerous ways of action are implicated, including the frequency and the intensity of insulin and other signaling pathway activation, the exertion of anti-inflammatory effects, the inhibition of nutrient absorption in the intestine (i.e., glucose or fatty acids) and, importantly, intestinal microbiome modification [2,8,9]. In fact, whereas the core microbiome of an individual is shaped early in life and remains relatively stable, its composition has been shown to be modulated by diet both rapidly and in the long-term [10,11] and has been proposed as a potential therapeutic target for a wide range of pathologies, including obesity and insulin resistance [12].

Diet supplementation with fish-derived oil containing omega 3 fatty acids has been shown to possess beneficial effects in diabetes and obesity [13]. Different types of extracts from fish have been tested for beneficial action on obesity and type 2 diabetes, such as tuna-derived extracts through boiling [14], Tapra fish-derived fish oil [15], or ethanol extracts from Masou salmon [16]. Yet, the contribution of fish extracts on the gut microbiome has not been investigated. In this study, Cod Powder (CP) and Fish Complex (FC), which consists of Codfish, Coalfish and Haddock, and contains higher concentrations of the health-promoting minerals Calcium, Phosphorus and Zinc, were tested individually or in combination. All of the aforementioned fish are abundantly harvested by fisheries around the world (many of which are certified as sustainable by the Marine Stewardship Council, a non-profit organization that secures sustainable fishing internationally) and are of considerable commercial interest.

Moreover, earlier studies have indicated that fish collagen may exert a beneficial effect on diet-induced obesity [17,18]. Collagen isolated from Skate (*Raja kenojei*) suppresses weight gain in diet-induced obese mice [19]. Notably, fish-derived collagen constitutes an intriguing alternative to mammalian collagen in terms of dietary supplementation, as the latter has been suggested to exert some side-effects, such as allergies and disease transmission, and marine collagen is produced by by-products of fisheries [20]. Thus, there has been an increasing interest in developing sustainable ways of producing collagen of marine origin with the potential for biomedical or biomaterial use [21,22]. The combination of fish collagen with fish-derived extracts on type 2 diabetes and obesity, as well as their impact on the microbiome has not been previously investigated.

Different animal models exist in order to study the effect of diet supplementation on obesity and type 2 diabetes, of which broadly used are diet-induced obesity models [23]. In the present study, a murine model of high-fat diet (HFD)-induced obesity and type 2 diabetes was utilized to examine the effects of fish-derived nutritional supplements on obesity and diabetes and determine changes in the gut microbiome.

## 2. Results and Discussion

### 2.1. Isolation of Fish Extracts and Nutritional Supplement Preparation

In order to investigate the potential beneficial role of fish supplements (Table 1) in high-fat induced obesity and insulin resistance, mice were fed a high-fat diet supplemented with 5% *w*/*w* of the respective supplement to be tested, for a period of 9 weeks. The 5% *w*/*w* supplementation was chosen as a sufficient simulation model for the average human supplement intake. Since fish supplements contain a substantial amount of protein, a 5% *w*/*w* soy protein supplemented diet was used as control to simulate a comparable protein intake. Detailed nutritional composition of supplemented diets is shown in Appendix A.

### 2.2. Fish Supplements Suppress Insulin Resistance in High-Fat Diet Mice Independently of Weight Gain

When animals are on a high-fat diet, they rapidly gain weight. In order to examine whether the selected supplements have an effect on the animal weight gain, mice following a high-fat diet supplemented with fish extracts had their weight monitored twice a week for 9 weeks. Animals that were fed the FC supplement had a similar weight accumulation with the HFD-control group (Figure 1A). On the other hand, CP supplementation caused a significant increase (*p* < 0.001) in animal weight gain during weeks 6–8 of the experiment (Figure 1B). This increase may be due to the higher energy content of the CP supplement compared to the FC. Interestingly, when CP was combined with FC, the elevated weight gain due to the former was reversed and a lowering tendency could be noted (*p* < 0.01) (Figure 1C). This could indicate a synergistic effect between the two extracts or could be the result of the lower concentration (2.5% *w*/*w*) of the CP supplement that resulted in weight gain (Figure 1B). The combination of FC with collagen had no effect on the weight change (Figure 1A). However, when CP was mixed with collagen, the increased weight accumulation was ameliorated (Figure 1B). That could be attributed to a possible anti-obesity effect of collagen, as has been previously reported [17,19], or to the smaller CP concentration.

Mice fed a HFD for sufficient time develop, in parallel to obesity, insulin resistance. Insulin resistance leads to impaired glucose tolerance, which can be measured with a glucose tolerance test (GTT). The glucose tolerance was monitored twice a week for two weeks and then once a week. Mice were fasted overnight, and glucose was measured in the blood before, and 30’, 60’ and 120’ minutes after intraperitoneal injection of dextrose. At approximately 3 weeks, the HFD-fed mice started exhibiting significantly delayed glucose clearance compared to the normal diet control group and the timing of the glucose intolerance appearance was the same among all groups (data not shown). Some of the tested nutritional extracts displayed a beneficial effect on the glucose tolerance at 8 weeks of supplementation, which lasted until the end of the experiment (9 weeks). Specifically, FC improved the glucose tolerance, whereas CP had no effect (Figure 2A,B). This may be attributed to omega 3 fatty acids, including DHA and EPA, which have anti-inflammatory properties and are present in the FC but not in the CP supplement. Interestingly, when CP was combined with FC, the aforementioned amelioration reappeared (Figure 2C). This is in agreement with the effect of the same supplements on weight gain. When FC was combined with collagen, however, no change in the glucose concentration was observed (Figure 2A). Nonetheless, the group of mice receiving CP mixed with collagen exhibited an improved glucose sensitivity (Figure 2B). Overall, FC seems to have a potent beneficial effect, and so does the combination of CP with collagen.

Increased weight gain is linked to abdominal fat accumulation. The visceral (epididymal for males, perigonadal for females) fat depot is thought to be an important (although not the sole) contributor to metabolic risk, including insulin resistance [24]. Perigonadal tissue was excised and its weight was measured right after animal sacrifice on day 66 of the experiment. All HFD-fed mice had considerably larger visceral adipose tissues compared to the normal diet control group. No significant change in adipose tissue weight was observed when mice were supplemented with the tested supplements (Figure 3A), although a tendency for reduced abdominal fat is observed in the groups with collagen-containing diet, which could be attributed to the proposed anti-obesity action of collagen [17,19].

While adipose tissue was initially regarded as an energy storage depot, it is now considered to be an active endocrine organ which regulates a number of homeostatic processes, such as metabolism, inflammation and blood pressure, mainly through adipokine and cytokine production [25]. The adipokine leptin is known as the “satiety hormone”, which regulates energy balance by inhibiting hunger, and has an insulin-sensitizing effect, while its structure resembles pro-inflammatory cytokines [26]. Leptin levels in white adipose tissue reflect the size of energy stores, therefore they appear higher in obese individuals [27]. Indeed, the control diet-fed mice exhibited a considerably lower leptin gene expression compared to the HFD control group, whereas it is noteworthy that the leptin gene expression levels follow the pattern of the corresponding adipose tissue weight. Both the FC and the CP groups displayed an elevated expression of leptin (Figure 3B), which could be attributed to their high energy content, possibly resulting in elevated leptin to induce satiety. Our results showed that the FC supplement led to an improved glucose tolerance, whereas CP did not, while it resulted in an increased body weight. Therefore, it is likely that the elevated leptin in the FC group exerted an insulin-sensitizing effect, whereas the CP group was leptin-resistant, which is common in obesity [27]. The combination of the two supplements had no significant effect on the leptin expression. When FC was mixed with collagen, there was no statistically significant change in leptin levels, but a tendency for a lower expression was noted. Yet, CP mixed with collagen promoted a statistically significant reduction of the leptin expression, approximating those of the control diet group, which is consistent with the improved glucose tolerance observed in this group of mice (Figure 3B). Altogether, our results showed that the FC may exert a leptin-sensitizing effect, leading to an improved glucose tolerance, while the combination of CP with collagen displayed an anti-obesity effect.

### 2.3. Fish Supplements Modulate the Gut Microbiome

Environmentally exposed tissues, like skin, mouth, gut, and vagina, are the most prominent body surfaces colonized by microorganisms. It is estimated that over 100 trillion bacteria colonize the distal gut, making the human gut one of the most complex ecosystems of the world, which also includes eukaryotic symbionts, altogether comprising the gut microbiome [28,29]. The gut is primarily colonized by symbiotic bacteria, being the main component of the intestinal microbiome that is known to affect health. Certain commensal bacteria are known to exert beneficial effects to the host, such as limiting colonization of potentially pathogenic microorganisms, providing nutrients, metabolizing undigested nutrients and even promoting the development and function of the immune system [30,31,32]. However, microbial imbalance and dysbiosis can be caused by external factors, such as antibiotic consumption, dietary changes, physical and psychological stress, resulting in gut enrichment with opportunistic microorganisms impairing the host’s wellness and homeostasis. The gastrointestinal microbial composition and host immune tolerance are subjected to a continuous interplay. Dysbiosis promotes the induction of host inflammatory responses and the onset of pathogenesis of a broad spectrum of diseases, such as Inflammatory Bowel Disease (IBD), celiac disease, obesity, colorectal cancer, autism spectrum disorder and it is even associated with the modulation of the host’s brain function and epigenome [33,34,35]. We, therefore, tested whether supplementation with fish-derived supplements alters the gut microbiome and affects its composition to reverse dysbiosis induced by a fat-containing diet.

For this purpose, mice were fed for 9 weeks with a high-fat diet supplemented with 5% *w*/*w* fish-derived supplements. Fecal samples were collected from the cecum and large intestine. DNA was extracted and a metagenome analysis was performed by 16S rRNA gene sequencing analysis. A Principal Component Analysis (PCoA) was conducted based on the identified OTUs, to assess genetic clustering of the different diet groups. Overall, 1,463,319 quality filtered sequences were generated that were assigned to 7137 unique OTUs. According to the PCoA, the first two axes (PCo1 and PCo2) accounted for 58.7% of the total variance, while the third (PCo3) explained 6.7% of the total variance. Overall, three distinct groups were formed; one including the lean diet-fed mice (ND), one including the groups supplemented with collagen-containing diets, being the FC+C and CP+C, and another one tight cluster containing the high-fat diet-fed, FC, CP and FC+CP groups. As expected, the PCoA showed significant differences between the lean (ND) and HFD groups. Interestingly, the groups supplemented with collagen-containing diets clustered together, in great distance from the other complexes, pointing to a differentiated bacterial niche within the gut microbiome (Figure 4A). Although FC and FC+CP promoted insulin sensitivity in high-fat diet-fed mice (Figure 2A,C), they exhibited no profound differences in the gut microbiome as demonstrated by the identified OTUs. The same applied for mice fed with CP (Figure 4A). α-diversity indices verified the clustering of groups according to the PCoA results. Independently of the index studied (Chao1, ACE, Shannon, Simpson, InvSimpson and Fisher) the ND-fed group was found the most diverse group in terms of species richness, counting on average 1303 unique OTUs, whereas groups supplemented with collagen-containing diets were proven to be the least diverse, counting on average 731 unique OTUs (Figure 4B). Comparison of the observed OTUs and the Chao1 index revealed that rare OTUs are present (singletons, doubletons). Taken together, the results indicate that rare taxa of gut microbes were displaced by ones favored by the supplement, which was also verified by the Shannon and Fisher diversity indices (Figure 4B).

The taxonomic classification of the identified OTUs revealed major differences, mainly among the genera comprising the ND-fed mice and the HFD-fed mice (Figure 5A). In particular, HFD-fed mice were represented by a larger fraction of uncultured bacteria compared to ND-fed mice, since nutrient diversity was restricted due to HFD and the effort to meet energy demands drove an increase in gut microbial biodiversity, but not an increase in beneficial microbial species [36]. In addition, collagen-containing supplements induced the colonization of *Bacteroides*, *Helicobacter, Alistipes* and *Odoribacter* genera, while they reduced the abundance of *Lachnospiraceae NK4A136 group* and *Ruminiclostridium 9* genera (Figure 5B). Although *Helicobacter pylori* is the most commonly detected species of the genus and associated with the induction of inflammation, non-*H. pylori* species were detected (Appendix A) [37,38]. *Bacteroides* is the predominant genus of the phylum Bacteroidetes present in the gut microenvironment [12]. Although members of that genus can colonize different parts of the body, such as blood, skin ulcers, urinary track and exhibit high antibiotic resistance, they are considered as beneficial organisms when restricted to the gut [39]. Indeed, in inflammatory bowel disease patients, *Bacteroides* has been found to be decreased and is associated with disease remission [40]. The *Alistipes* genus is also member of the Bacteroidetes phylum, yet it is underrepresented in the gut microbiome, and it is usually found in healthy individuals. Contradicting evidence on its implication in health and disease pathogenesis exist; their presence is associated with mental illness, yet protective effects have been demonstrated in colitis, liver fibrosis, cancer immunotherapy, and cardiovascular disease [41]. In addition to their role in actively competing pathogenic bacteria, gut microorganisms can contribute in reduced intestinal inflammation through the production of metabolites such as butyrate and short chain fatty acids (SCFA). Members of the genus *Odoribacter* are known to produce acetate, propionate and butyrate, all known to possess anti-inflammatory and anti-diabetic actions [42], and ameliorate intestinal inflammation through the production of SCFA. Not only collagen-containing supplements but also fish complex-containing supplements could potentially contribute to a reduced intestinal inflammation. In parallel to the induction of beneficial bacterial genera, collagen-containing supplements reduced the abundance of *Lachnospiraceae NK4A136 group* and *Ruminiclostridium 9* genera, which are both butyrate producers (Figure 5B). Overall, this could be explained by the fact that intestinal colonization is a constant competitive process of microbes competing for nutritional resources, driving the reduction of certain species over the dominance of other more adaptive ones (summarized in Appendix A).

## 3. Materials and Methods

### 3.1. Fish Extracts

Collagen (fish skin protein powder) from the Seagarden company, Avaldsnes, Norway was extracted from Codfish skin (*Gadus morhua*). The skin had an abundant amount of collagen protein. Collagen was extracted by acid hydrolysis and digested using a mixture of proteases (180 mL Papain from Enzybel and 180 mL Neutrase from AB Enzymes 360 mL/4000 kg cod skin). Prior to extraction, the fish skin was pretreated with water to remove salt and blood debris, followed by alkali (NAOH 45%, 10 L/4000 kg cod skin) treatment to remove the non-collagenous proteins. Following the extraction of collagen with acetic acid, the preparation was filtered using active carbon and then concentrated by evaporation to around 45% humidity. The extract was then dried using the spray-dry technology (GEA dryer MS 850, GEA, Dusseldorf, Germany). Nutritional, mineral and microbiological analyses for the supplements were commercially performed by ALS Laboratory, Oslo, Norway, SynLabs, Stjørdal Norway, and SLab, Stord, Norway. The results are shown in Table 2, Table 3 and Table 4, respectively.

Cod Powder was prepared from 100% Cod and Fish Complex was a mixture of 50% Cod, 40% Coalfish and 10% Haddock. Codfish (*Gadus morhua*), Coalfish (*Pollachius virens*), Haddock fish (*Melanogrammus aeglefinus*) and natural antioxidants (tocopherol-rich extracts) generated by the Seagarden company, Avaldsnes, Norway, were cooked in steam for 40 min at different controlled temperatures that preserved the fish nutritional and organoleptic properties without degrading its bioactive constituents. Preparations were then dried using the air-drying process where the fish side stream is cooked at a temperature that secures the quality of the products and minimizes the loss of protein and taste active components. After cooking, dehydration was carried out through continuous air-drying (CPM Wolverine proctor, Hosham, PA, USA). The temperature was strictly controlled to minimize flavor degradation. The cooked, dehydrated product was then micro-milled (Hosokawa Alpine Aktiengesellschaft, Augsburg, Germany) to fine powder under 106 micrometers, where 80% of the particles were less than 106 micrometers. Nutritional analyses for Cod Powder and Fish Complex are also shown in Table 2 and Table 3.

All supplements were provided to the animals as 5% supplement embedded to the diet, which was a High Fat Diet preparation (DIO rodent purified Diet W/60% energy from fat, catalog number PF4051/D, purchased by Test Diets, IPS Product Supplies Ltd, London, UK).

### 3.2. Animal Maintenance-Protocols

The C57BL/6 mice were maintained in a 12h day/night cycle and 21–23°C conditions prior to treatments in a pathogen-free animal facility in the Medical School of the University of Crete, Heraklion. All procedures were conducted in compliance with protocols approved by the Animal Care Committee of the University of Crete, School of Medicine (Heraklion, Crete, Greece) and the Veterinary Department of the Region of Crete under license number 269904 (Heraklion, Crete, Greece). For the diet-induced obese phenotype, 5 (6 weeks old) female mice per group were fed a High-Fat Diet for 9 weeks. Each group was supplemented with 5% *w*/*w* of the corresponding fish supplement. A total of 5% *w*/*w* of soy protein was used as control supplement since the major source of protein in rodent grain-based diets is soy protein. Specifically, one group of mice was fed a standard diet supplemented with 5% *w*/*w* soy protein and another one was fed a high-fat diet also supplemented with 5% *w*/*w* soy protein. For the glucose tolerance test (GTT) the mice were fasted overnight and then glucose measurements were conducted to determine the basal glucose levels referred as timepoint 0. Then, the mice were injected intraperitoneally with sterile 35% dextrose (100 μL dextrose/35 g mice) and the blood levels of glucose were determined 30, 60 and 120 min post injection. All glucose measurements were conducted using the TRUEresult^®^ Twist meter (Nipro diagnostics. Fort Lauderdale, FL, USA).

### 3.3. RNA Extraction, cDNA Synthesis and Quantitative PCR

Epididymal white fat tissue (eWAT) was collected immediately after animal sacrifice, the weight was measured, and the tissue was stored at −80 °C for further analyses. The total eWAT was homogenized using the Trizol^TM^ reagent, (Termo Fisher, Carlsbad, CA, USA), on a mechanical homogenizer. RNA extraction was performed according to the manufacturer’s instructions. Then, 500 ng of the total RNA was reverse transcribed using the PrimeScript^TM^ RT Master Mix (Perfect Real Time) (RR036A, TaKaRa, Tokyo, Japan) following the manufacturer’s instructions. Two-step quantitative PCR was performed in technical duplicates using the Kapa SyBr^®^ Fast Universal kit (KK4618, Merck, Darmstadt, Germany) on a 7500 Fast Real-Time PCR Instrument (Applied Biosystems^®^, 4351106, Foster City, CA, USA) with 96-well Block Module as follows: start step 95 °C for 3 min and then 40 cycles of 95 °C for 10 s and 60 °C for 30 s. Primers were designed using the primer designing tool of the NCBI database selecting the gene sequence also from the NCBI database and choosing to span an exon–exon junction, 60 °C Tm point and PCR products between 70 and 200 bps. *Ppia* mRNA was utilized as internal control gene. Primer sequences were designed as follows: *Ppia* Forward 5′-ATGGTCAACCCCACCGTGT-3′ and reverse 5′-TTCTGCTGTCTTTGGAACTTTGTC-3′ amplifying 102 bps, *Leptin* forward 5′-TCACACACGCAGTCGGTATC-3′ and reverse 5′-GCACATTTTGGGAAGGCAGG-3′ amplifying 148 bps. The data analysis was conducted using mRNA levels expressed as relative quantification (RQ) values, which were calculated as RQ = 2^DDCt^, where DCt is (Ct (gene of interest) minus Ct (housekeeping gene)).

### 3.4. Microbiome Extraction and Sequencing

Fecal samples were collected from the large intestine and cecum. The amount of 200 mg of stool sample were used to perform DNA extraction using the NucleoSpin DNA stool kit (740472.50, Macherey-Nagel, Duren, Germany) according to the manufacturer’s instructions. An amplicon metabarcoding analysis was conducted by amplifying and sequencing the V3 to V4 hypervariable regions of the 16S rRNA gene (≈460 bp). Libraries were constructed according to Illumina’s protocol for 16S post-genomic analyses described in “Illumina’s 16S Metagenomic Sequencing Library Preparation” (15044223 B) manual. For the amplification of V3 to V4 regions primers were selected from Klindworth et al., 2013 [43] and were modified by adding an Illumina adapter sequence at the 5′ end of each primer. The size and quality of the generated library were evaluated on the Fragment Analyzer system (Agilent Technologies Inc., Santa Clara, CA, USA) using the DNF-477-0500 kit. Sequencing was conducted on a MiSeq platform (Illumina Inc., San Diego, CA, USA) using the MiSeq^®^ reagent kit v3, (Illumina Inc., San Diego, CA, USA). For the identification of the prokaryotic load bioinformatic an analysis was performed using the Qiime2 pipeline [44] (Quantitative Insights into Microbial Ecology). Quality-trimmed (minimum quality score of 28) and non-chimeric reads were assigned to OTUs (Operational Taxonomic Units) at 99% nucleotide homology against the SILVA 132 database, Max Plank Institute, Bremen, Germany, using the open-reference method and the VSEARCH tool (Versatile open Source tool for metagenomics) [45]. Phylogenetic relationships of the identified OTUs and samples were constructed in R version 3.6.0 [46], using the phyloseq R [47], ampvis2 [48], and ggplot2 [49] R packages. Barplots depicting OTU counts are normalized to 100% as abundance estimations within each sample.

### 3.5. Statistical Analysis

Data are presented as mean ± SD. A statistical analysis was performed using Graphpad Prism7.0. D’Agostino–Pearson, Shapiro–Wilk and KS tests were used to evaluate normality. As the samples did not follow a Gaussian distribution, a Mann–Whitney *t*-test was performed to test the statistical analysis of each supplement to the control diet. A 2-way Anova was performed to confirm the statistical significance of the results. Differences with a *p* value < 0.05 are considered significant (* indicates *p* < 0.05, ** indicates *p* < 0.01, *** indicates *p* < 0.001).

## 4. Conclusions

Herein, we demonstrate that fish-derived nutritional supplements suppress glucose intolerance in a model of high-fat diet-induced obesity and type 2 diabetes in mice and specifically nutritional supplements containing fish complex, fish complex combined with cod powder and cod powder combined with collagen. In addition, nutritional supplements containing fish-derived collagen modulated the gut microbiome in obese mice, inducing colonization of beneficial bacteria, known to have beneficial properties in suppressing metabolic inflammation and diabetes. Thus, our results suggest that fish-derived extracts may suppress type 2 diabetes and that this action may be partly due to collagen-induced modulation of the gut microbiome.

## Figures and Tables

**Figure 1 marinedrugs-19-00268-f001:**
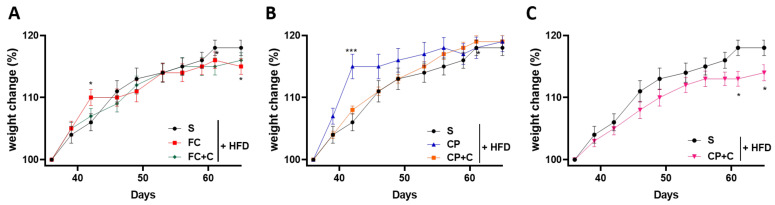
The effect of fish-derived dietary supplements on animal weight gain during high-fat diet administration. (**A**–**C**) Total body weight change (%). Graphs represent mean ± SD and Mann–Whitney *t*-test was performed. * *p* < 0.05, *** *p* < 0.001.

**Figure 2 marinedrugs-19-00268-f002:**
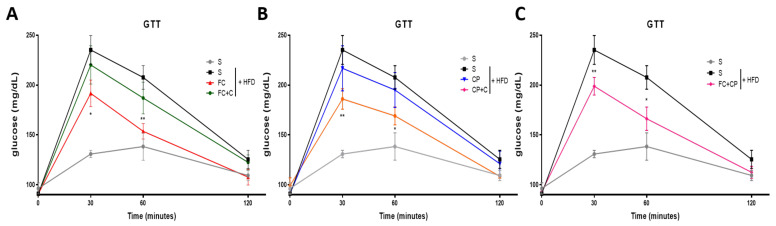
The effect of fish-derived dietary supplements on glucose tolerance in high-fat diet-fed mice on week 8. (**A**–**C**) Mice were fed with the indicated nutritional supplements and then subjected to a glucose tolerance test on day 53. Graphs represent mean ± SEM and Mann–Whitney *t*-test was performed. * *p* < 0.05, ** *p* < 0.01.

**Figure 3 marinedrugs-19-00268-f003:**
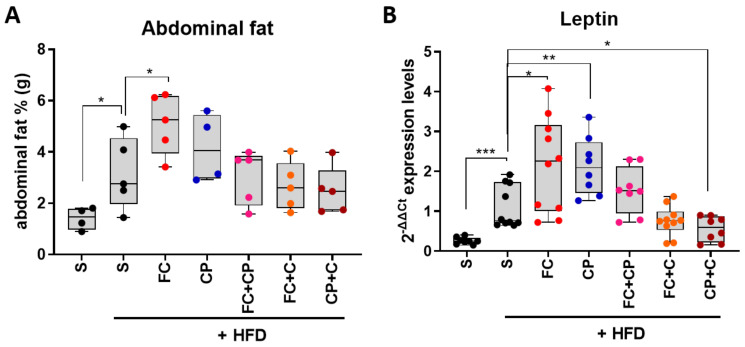
The effect of fish-derived dietary supplements on abdominal fat and leptin expression in abdominal fat tissue of high-fat diet-fed mice. Mice were fed with the indicated nutritional supplement (**A**). Abdominal fat in grams normalized to total body weight was measured (**B**). Leptin expression levels in abdominal fat tissue were quantified using real-time PCR. A Mann–Whitney *t*-test was performed. * *p* < 0.05, ** *p* < 0.01, *** *p* < 0.001.

**Figure 4 marinedrugs-19-00268-f004:**
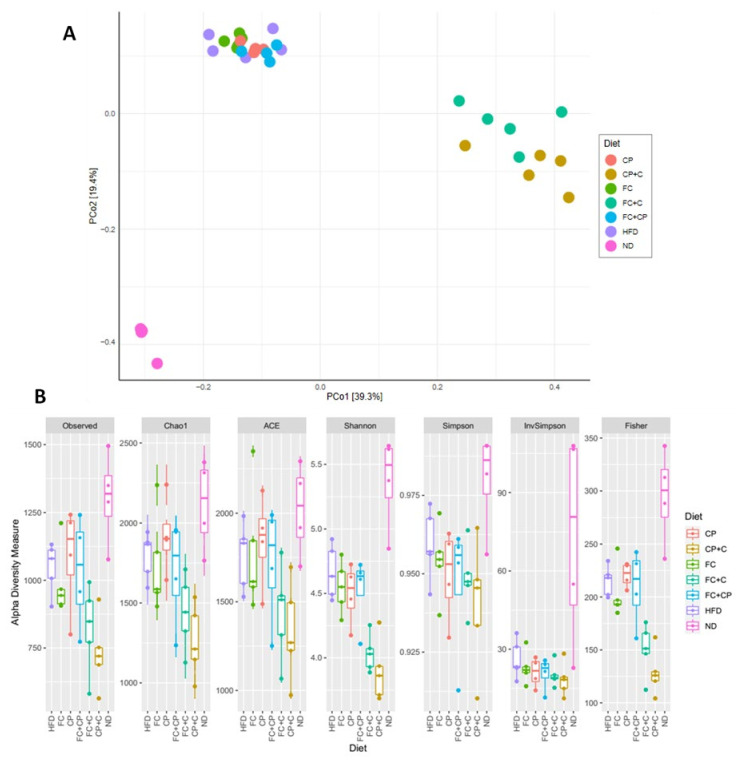
The effect of fish-derived dietary supplements on gut bacterial diversity. (**A**) PCoA analysis and (**B**) Alpha Diversity analysis; HFD: high fat diet-fed; FC: Fish Complex; CP: Cod Powder; FC + CP: Fish Complex + Cod Powder; FC + C: Fish Complex + collagen; CP + C: Cod Powder + collagen; ND: Normal Diet.

**Figure 5 marinedrugs-19-00268-f005:**
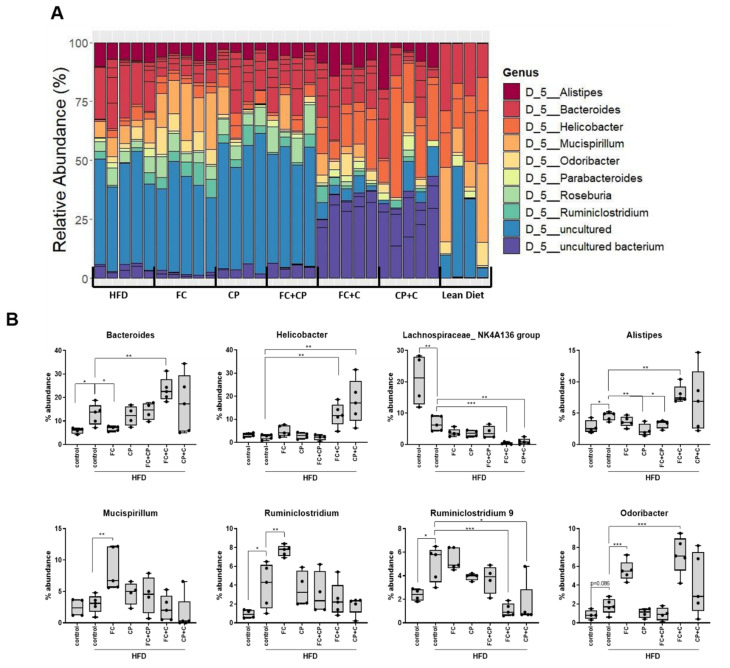
The effect of fish-derived dietary supplements on gut microbiome top genera composition. (**A**) Normalized barplots of genera abundances (%) based on the identified OTUs. (**B**). Boxplots representing the abundance of different genera following diet supplementation. Mann–Whitney *t*-test was performed. HFD: high-fat diet-fed; FC: Fish Complex; CP: Cod Powder; FC + CP: Fish Complex + Cod Powder; FC + C: Fish Complex + collagen; CP + C: Cod Powder + collagen; Lean Diet: Normal Diet. * *p* < 0.05, ** *p* < 0.01, *** *p* < 0.001.

**Table 1 marinedrugs-19-00268-t001:** List of the different diet groups and the composition of the supplements used.

Diet	Supplement	Composition
Standard	S	Soy protein
High-fat	S	Soy protein
	FC	White fish powder
	CP	Cod powder
	FC + CP	50%/50% White fish powder + cod powder
	FC + C	50%/50% White fish powder + fish skin protein powder
	CP +C	50%/50% Cod powder + fish skin protein powder

**Table 2 marinedrugs-19-00268-t002:** Nutritional analysis of the different supplements.

Chemical Composition	Unit			Collagen	Fish Complex	Cod Powder
Energy	Kcal/100 g extract			389	304	385
	KJ			1634	1274	1620
Total solids	g/100 g extract			≥92	94	94
Protein	g/100 g extract			≥95	66	87
Ash	g/100 g extract			<2	22	7
Salt	g/100 g extract			<1	1.2	2
Fat	g/100 g extract			<0.5	4.4	4
		Saturated			1.15	0.9
		Monounsaturated			2.08	0.9
		Polyunsaturated			1.74	1.8
			Omega-6		0.2	0.2
			Linoleic acid		0.08	0.11
			Omega-3		1.5	1.6
			DHA		0.87	0.97
			EPA		0.43	0.49

**Table 3 marinedrugs-19-00268-t003:** Mineral content analysis of the different supplements.

Minerals	Unit	Collagen	Fish Complex	Cod Powder
Calcium	mg/100 g	not detected	6810	934
Phosphorus	mg/100 g	not detected	4890	993
Sodium	mg/100 g	not detected	730	800
Potassium	mg/100 g	not detected	571	1360
Magnesium	mg/100 g	not detected	215	120
Iron	mg/100 g	not detected	20.8	1.1
Zink	mg/100 g	not detected	7.85	2.71
Iodine	mg/100 g	not detected	0.43	0.59
Selenium	mg/100 g	not detected	0.22	0.12
Vitamin B12	mg/100 g	not detected	0.0067	0.0069

**Table 4 marinedrugs-19-00268-t004:** Microbiological analysis of the supplements.

Microorganism	Unit	Collagen	Fish Complex	Cod Powder
Total plate count	Max CFU/g	<1000	50,000	50,000
*Enterobacteriaceae*	Max CFU/g	<100	100	100
*Escherichia coli*	Max CFU/g	Negative in 1 g	Negative in 1 g	<10
*Salmonella*	Max CFU/g	Negative in 25 g	Negative in 25 g	Negative in 25 g
Mold and yeast	Max CFU/g	<200	500	1000
Sulphite-Reducing Clostridia	Max CFU/g	<100	<100	-
*Listeria monocytogenes*	Max CFU/g	Negative in 25 g	Negative 25 g	Negative in 25 g

CFU: Colony Forming Units; (-) not detected.

## Data Availability

The data presented in this study are available on request from the corresponding author and can be made publicly available.

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
