# Peer review of "Diet Supplementation with Fish-Derived Extracts Suppresses Diabetes and Modulates Intestinal Microbiome in a Murine Model of Diet-Induced Obesity"

_marinedrugs, 2021, doi:10.3390/md19050268_

Round 1
Reviewer 1 Report
In this manuscript, the authors described the effect of fish-derived extracts on anti-diabetes activity. They reported that dietary supplementation with fish-derived extracts could improve glucose intolerance and enhance leptin expression. Besides, they found that collagen extracted from fish skin could regulate the gut microbiome composition so that suppress diet-induced diabetes.
Comments
- Overall, the authors investigated the biological effects of fish complex powder, Cod powder, and collagen. According to the materials, the fish complex powder consists of several fish nutrients, including those from Cod. The purpose of studying the fish complex is a little confusing. The introduction and discussion about why study the fish complex need to be added. Besides, the purpose of collagen added also needs to be discussed. It’s should be more informative if there is a group fed only with collagen.
- In figure 1, what did the x-axis indicate? The mice were treated for 9 weeks, but this figure only showed weight change in 4 weeks. In line 200, the authors described that “the mice were fed for 10 weeks with a high-fat diet supplemented with 5% w/w…”. Is this a different set of an experiment? The methods should be described clearly and consistently.
- What’s the time point of the GTT measurement showed in figure 2? It should be clarified.
- The 16s rRNA sequencing results are interesting, however, the sample collection is not clear. In this study, the fecal samples were collected from the colon, caecum, and terminal ileum. The alteration of bacterial composition in different segments of the gut has been reported (PMID: 25887695). In this case, how to make sure the collected sample represented the gut microbiome composition in this mouse? In other words, how to make sure the samples are homogenous?
Author Response
We thank the Reviewer for the constructive comments. Below is a point by point response to the comments.
Comment 1.1: Overall, the authors investigated the biological effects of fish complex powder, Cod powder, and collagen. According to the materials, the fish complex powder consists of several fish nutrients, including those from Cod. The purpose of studying the fish complex is a little confusing. The introduction and discussion about why study the fish complex need to be added. Besides, the purpose of collagen added also needs to be discussed. It should be more informative if there is a group fed only with collagen.
- Response 1.1: We thank the reviewer for the insightful comment. We have included a section in the introduction supporting the purpose of studying fish complex and cod powder in lines 71-76 and 79-84. The effects of collagen as dietary supplement in the context of obesity has been previously tested and is mentioned in lines 77-84 in the introduction. Therefore, our analysis focused on investigating potential synergistic effects that could arise from the combination of collagen with other fish-derived extracts, which has not been studied before.
Comment 1.2: In figure 1, what did the x-axis indicate? The mice were treated for 9 weeks, but this figure only showed weight change in 4 weeks. In line 200, the authors described that “the mice were fed for 10 weeks with a high-fat diet supplemented with 5% w/w…”. Is this a different set of an experiment? The methods should be described clearly and consistently.
- Response 1.2: We apologize to the reviewer for the confusion. The x-axis in figure 1 indicates the days post the initiation of the experiment. As mice were 6 week old and during the first weeks gained weight not only due to the high fat diet but also because of the developmental process, we presented the results starting on week 4 where the effect of developmental growth was not anymore present. We have also replaced “10 weeks” with “9 weeks” in line 212, which was an error.
Comment 1.3: What’s the time point of the GTT measurement shown in figure 2? It should be clarified.
- Response 1.3: We thank the reviewer for pointing out this important omission. It has been clarified in the legend of figure 2 (line 147) that the GTT measurements were executed on day 53 (week 8), on which the obese phenotype had been fully established.
Comment 1.4: The 16s rRNA sequencing results are interesting, however, the sample collection is not clear. In this study, the fecal samples were collected from the colon, caecum, and terminal ileum. The alteration of bacterial composition in different segments of the gut has been reported (PMID: 25887695). In this case, how to make sure the collected sample represented the gut microbiome composition in this mouse? In other words, how to make sure the samples are homogenous
- Response 1.4: We thank the reviewer for the comment. The fecal samples were all collected and processed by the same person and at the same time to minimize collection bias. In order to keep samples as representative as possible, fecal samples were first collected and homogenized in a test tube and then 200 mg of feces were processed further for DNA extraction and sequencing. Regarding the location of the collection, you are correct indeed that microbial composition is highly dependent on the intestinal location. Therefore, we opted to collect samples from the cecum and colon as it has been shown that it exhibits the greatest similarity concerning genus and phylum microbial composition (PMCID: PMC4369366 ,PMCID: PMC3792069 ). In order to avoid confusion, “terminal ileum” has been erased from lines 213 and 363 as the term was used only to describe the fact that when cecum was isolated from mice there was always a small portion of the terminal ileum was also present when excising the cecum, but too small and at the same proportion to affect the results.
Reviewer 2 Report
Dear Authors,
Please see my comments in the attached file

Author Response
We thank the Reviewer for the constructive comments. Below is a point by point response to the comments.
Comment 2.1: Line 28 is it a mix of this? please check this information
- Response 2.1: We apologize to the reviewer for the confusion. The supplement was a mix of cod powder and fish complex. The description in the abstract has been modified, so that the information would be clearer.
Comment 2.2: Line 42 erase “by WHO”
- Response 2.2: “By WHO” was deleted in line 43. Thank you.
Comment 2.3: Line 44 please add a reference
- Response 2.3: We thank the reviewer for the comment. Reference (PMID: 31189627) was added to line 45.
Comment 2.4: Line 70. it should be interesting to add information about the use of collagen isolated from marine species as raw-materials for development of biomaterials for biomedical application eg:
1) Extraction and Characterization of Collagen from Elasmobranch Byproducts for Potential Biomaterial Use
2) Collagen from Atlantic cod (Gadus morhua) skins extracted using CO2 acidified water with potential application in healthcare
- Response 2.4: We thank the reviewer for the interesting suggestion. “Additionally, there has been an increasing interest in developing sustainable ways of producing collagen of marine origin with the potential for biomedical or biomaterial use” has been added in lines 83-84. Also the suggested references have been added in the text, line 84.
Comment 2.5: Line 88 table 1. it should be recommended to add an acronym name to understand better, please check the abstract and correct the name eg. 1) FishC 2) CodP 3)FishC+CodP 4) FishC+Coll 5) CodP+Coll
- Response 2.5: We thank the reviewer for the suggestion. The following acronyms have been used in table 1 as well as in all manuscript text 1) FC (Fish complex), 2) CP (Cod Powder), 3) FC+CP (Fish Complex + Cod Powder), 4) FC+C (Fish Complex + Collagen), 5) CP+C (Cod Powder + Collagen). Also table 1 has been replaced with one that includes more detailed information (line 101) in which type of diet, acronyms and diet composition are clearly stated.
Comment 2.6: Line 109 figure 1. The significant p values (p<0.05) should be incorporated in the text
- Response 2.6: We thank the reviewer for the comment. p<0.001 and p<0.01 were added in lines 110 and 114 respectively.
Comment 2.7: Line 189. please add a recent reference eg:
Vibrio diabolicus challenge in Bathymodiolus azoricus populations from Menez Gwen and Lucky Strike hydrothermal vent sites
Gene expression study in Bathymodiolus azoricus populations from three North Atlantic hydrothermal vent sites
- Response 2.7: Thank you for noting the outdated references. We have replaced them with more recent ones, including the ones you suggested in line 196.
Comment 2.8: Line 206 Please, add the value obtained of each cluster, namely proportion of variance obtained
- Response 2.8: We apologise to the reviewer for the confusion. The proportions of variance obtained for the first three axes were added in the revised manuscript in lines 217-219.
Comment 2.9: Line 242 it is missing a reference
Response 2.9: Thank you for your comment. We have added two references in line 261.
Comment 2.10: Line 242-266. I suggested to convert this information into a table that defined the bacteria (bacteroides) and related problem (eg. inflammation) location (tissue) and reference [11].
Response 2.10: We have generated a table that summarizes the information presented in the text. This table is included as Supplemental table 2.
Comment 2.11: Line 278 which is the composition of proteases cocktail?
- Response 2.11: We have now included the composition of the proteases cocktail which was “180 mL Papain from Enzybel and 180 mL Neutrase from AB Enzymes 360 mL/4000 kg cod skin”. This information is inserted in lines 293-294.
Comment 2.12: Line 280 it is missing the concentration.
- Response 2.12: We have now included the concentration which was “NAOH 45 %, 10 L/4000 kg cod skin”, line 295.
Comment 2.13: Line 283 it is missing the equipment name
- Response 2.13: “(GEA dryer MS 850)” was inserted in line 299.
Comment 2.14: Line 283 the procedure of these analyses is missing.
- Response 2.14: We have now included the information on the analyses , which were commercially performed by ALS Laboratory, Norway, SynLabs, Norway, and SLab, Norway. The information is included in lines 299-301.
Comment 2.15: Line 285 what was the ratio between the components
- Response 2.15: We thank the reviewer for the comment. “Cod Powder and Fish Complex were prepared from Cod and a mixture of Cod, Coalfish and Haddock, respectively“ was substituted with “Cod Powder was prepared from 100% Cod and Fish Complex was a mixture of 50 % Cod, 40 %Coalfish and 10 % Haddock.” in line 302-303.
Comment 2.16: Line 290 please, explain this information
- Response 2.16: We thank the reviewer for the comment. We have included information on the procedures In lines 307-314 as follows: “Preparations were then dried using air-drying process where the fish side stream is cooked at a temperature that secures the quality of the products and minimizes the loss of protein and taste active components. After cooking, dehydration was carried out through continuous air-drying techniques (CPM Wolverine proctor). The temperature was strictly controlled to minimize flavor degradation. The cooked, dehydrated product was then micro-milled (Hosokawa Alpine Aktiengesellschaft) to fine powder under 106 micrometers, where 80% of particles were less than 106 micrometers.“
Comment 2.17: Line 291 using which equipment?
- Response 2.17: The equipment (Hosokawa Alpine Aktiengesellschaft) has been added in line 312.
Comment 2.18: Line 292 erase “in size”
- Response 2.18:. “in size” has been erased. Thank you.
Comment 2.19: Line 299 table 3 The units should be in the same range
- Response 2.19: Vitamin B12 units were modified from ug/100g to mg/100g to match the units of the remaining molecules.
Comment 2.20: Line 300 table 4 The name of species should be in italic form
- Response 2.20: Thank you for pointing this out. The names of the species in table 4 were changed to italics.
Comment 2.21: Line 303 which was the number of mice? and ratio of male/female and also age of the animals
- Response 2.21: Each group consisted of 5 female mice, which is mentioned in line 331.
Comment 2.22: Line 327-330 which is the bp of product expected? Complete the information. How did you design the primers and how do you choose the sequence (NCBI database? for mice?
- Response 2.22: We thank the reviewer for the comment. We have included the information according to the suggestion in lines 352-354. The primers for the real time PCR experiment were designed using the primer designing tool of NCBI database selecting the gene sequence also from the NCBI database and choosing to span an exon-exon junction, 60 C Tm point and PCR products between 70 and 200 bps. Target sequence for the reference gene was 102 bps (inserted in line 357) and target sequence for leptin was 148 bps (inserted in line 359). We also apologise to the reviewer for our mistake as reference gene 18S in lines 355 and 366 was corrected to Ppia (Peptidylprolyl Isomerase A).
Comment 2.23: Line why did you use a nonparametric test? Please complete the information
- Response 2.23: We thank the reviewer for the comment. Non-parametric test was used as normality test was performed and samples did not follow a Gaussian distribution. This information is included as follows: “ D’Agostino & Pearson, Shapiro Wilk and KS tests were used to evaluate normality. As samples did not follow a Gaussian distribution a Mann–Whitney t-test was performed to test statistical analysis of each supplement to the control diet”, lines 386-388.
Reviewer 3 Report
The work is interesting, but the methods description is unclear.
Please consider the listing of all groups of rats in a single plot for body weight gain over the course of the experiment and glucose tolerance. Why was the analysis of variance not used in the statistical analysis?
When describing the methods, the authors stated that a control group was fed HF diet with addition of 5% of soy protein. In the sentence "5% w / w of soy protein was used as control supplement since the major source of protein in rodent diet is soy protein." (page 10, lines 310-311), please mention that it is for grain based diets. In purified diets, casein is the main source of protein for rodents.
The paper provides the chemical characteristics of fish derived supplements; as the energy value as well as the content of other nutrients varies, please provide information on the dietary intake for each group of mice.
The description of the control groups is somewhat unclear, was the control group fed a standard diet or a high fat diet? Is the second control group fed a high-fat supplemented with 5% soy protein? Please consider the change of marking of these groups in Figure 3.
Author Response
We thank the Reviewer for the constructive comments. Below is a point by point response to the comments.
Comment 3.1: Please consider the listing of all groups of rats in a single plot for body weight gain over the course of the experiment and glucose tolerance. Why was the analysis of variance not used in the statistical analysis?
- Response 3.1: We apologize to the reviewer for the confusing graphs. Show all groups of supplements in one graph as shown in the attached file, we believe is difficult for the reader as lines overlap. Instead, figures 1 and 2 were combined in pairs, as we believe that these graphs are easier to read. Although there are different dietary supplements tested in each graph, each supplement is compared with the control (Soy).(please see revised Figures 1 and 2). We have also repeated statistical analysis using 2 way Anova resulting in similar results. This information is introduced in the Materials and Methods section line 389.
Comment 3.2: When describing the methods, the authors stated that a control group was fed HF diet with addition of 5% of soy protein. In the sentence "5% w / w of soy protein was used as control supplement since the major source of protein in rodent diet is soy protein." (page 10, lines 310-311), please mention that it is for grain based diets. In purified diets, casein is the main source of protein for rodents.
- Response 3.2: We thank the reviewer for pointing out the need for this clarification. It is now mentioned in line 334-336 that soy protein is the major source of protein specifically in grain-based diets, as suggested.
Comment 3.3: The paper provides the chemical characteristics of fish derived supplements; as the energy value as well as the content of other nutrients varies, please provide information on the dietary intake for each group of mice.
- Response 3.3: We thank the reviewer for the insightful comment. Table 1 in supplementary material was added containing information about the intake of energy, carbohydrates, protein and fat for each diet group per 100g of diet.
Comment 3.4: The description of the control groups is somewhat unclear, was the control group fed a standard diet or a high fat diet? Is the second control group fed a high-fat supplemented with 5% soy protein? Please consider the change of marking of these groups in Figure 3.
- Response 3.4: We apologise to the reviewer for the confusion. One control group was fed a standard diet supplemented with 5% w/w soy protein, whereas the other control group was fed a high-fat diet also supplemented with 5% w/w soy protein. The marking of the control groups was changed from “control” to “S” or “S + HFD” (S for soy) throughout the manuscript (please see also response to Reviewer 1).

Round 2
Reviewer 1 Report
All of the queries raised in my report have been considered in full detail and the revisions in the new manuscript address these completely. There are no further issues to address. I would like to thank the authors for their detailed and professional response, which made it a pleasure to review their work.
Reviewer 3 Report
Most of the comments were taken into account.